# Soluble Epoxide Hydrolase Inhibition Attenuates Proteinuria by Alleviating Renal Inflammation and Podocyte Injuries in Adriamycin-Induced Nephropathy

**DOI:** 10.3390/ijms251910629

**Published:** 2024-10-02

**Authors:** Qingyu Niu, Ziyu Guo, Yaoxian Liang, Li Zuo

**Affiliations:** Department of Nephrology, Peking University People’s Hospital, Beijing 100044, China; qingyuniu1992@163.com (Q.N.); ziyuguo@pku.edu.cn (Z.G.); zuoli@bjmu.edu.cn (L.Z.)

**Keywords:** soluble epoxide hydrolase, proteinuria, inflammation, primary glomerular disease, adriamycin-induced nephropathy

## Abstract

Soluble epoxide hydrolase (sEH) has previously been demonstrated to play an important part in kidney diseases by hydrolyzing renoprotective epoxyeicosatrienoic acids to their less active diols. However, little is known about the role of sEH in primary glomerular diseases. Here, we investigated the effects of sEH inhibition on proteinuria in primary glomerular diseases and the underlying mechanism. The expression of sEH in the renal tubules of patients with minimal change disease, IgA nephropathy, and membranous nephropathy was significantly increased. Renal sEH expression level was positively correlated with the 24 h urine protein excretion and negatively correlated with serum albumin. In the animal model of Adriamycin (ADR)-induced nephropathy, renal sEH mRNA and protein expression increased significantly. Pharmacological inhibition of sEH with AUDA effectively reduced urine protein excretion and attenuated renal pathological damage. Furthermore, sEH inhibition markedly abrogated the abnormal expressions of nephrin and desmin in glomerular podocytes induced by ADR. More importantly, AUDA treatment inhibited renal NF-κB activation and reduced TNF-α levels in rats with ADR-induced nephropathy. Overall, our findings suggest that sEH inhibition ameliorates renal inflammation and podocyte injury, thus reducing proteinuria and exerting renoprotective effects. Targeting sEH might be a potential strategy for the treatment of proteinuria in primary glomerular diseases.

## 1. Introduction

Proteinuria is not only a common sign of glomerular diseases but also an independent risk factor for the progression of chronic kidney disease (CKD). Accumulating evidence suggests that inhibition of proteinuria can delay the progression of CKD [1]. Several pharmacological agents have been employed for the treatment of proteinuria, such as renin–angiotensin system inhibitors, sodium-glucose cotransporter-2 inhibitors, non-steroidal mineralocorticoid receptor antagonists, glucocorticoids, or immunosuppressive medications. Nevertheless, proteinuria cannot be sufficiently controlled and kidney function deteriorates progressively in some patients.

Multiple pathways are involved in the onset of proteinuria, among which podocyte dysfunction plays a pivotal role [2]. Podocytes are essential for maintaining the integrity and normal function of the glomerular filtration barrier. Various pathogenic factors, such as inflammation and oxidative stress, can cause the loss of nephrin in podocyte slit diaphragms and the disorganization of the actin cytoskeleton, resulting in impaired podocyte function and elevated proteinuria [3,4]. Therefore, protecting against podocyte injury is an effective strategy to reduce proteinuria and attenuate renal damage.

Epoxyeicosatrienoic acids (EETs) are the cytochrome P450 epoxygenase metabolites of arachidonic acid. Once formed, EETs act in an autocrine or paracrine manner to elicit anti-inflammatory, anti-apoptotic, and anti-oxidative actions that contribute to kidney protection [5,6]. However, EETs can be rapidly metabolized by soluble epoxide hydrolase (sEH), causing the loss of their protective effects. The sEH is widely distributed in the renal proximal tubular epithelial cells. Many previous studies, including ours, have demonstrated that genetic sEH deficiency, or its pharmacological inhibition, enhances EET levels and thus attenuates inflammation, fibrogenesis, podocyte injuries, and proteinuria [6,7,8,9,10]. It has been reported that sEH inhibition could protect against acute kidney injury, diabetic nephropathy, lupus nephritis, hypertensive renal damage, tubular epithelial–mesenchymal transition, and renal fibrosis [7,8,11,12,13,14,15]. However, relatively little is known about the role of sEH in primary glomerular diseases.

In the present study, we investigated renal sEH expressions in patients with primary glomerular diseases, including minimal change disease (MCD), IgA nephropathy (IgAN), and membranous nephropathy (MN). Furthermore, we studied the effects of sEH inhibition on proteinuria and podocyte injuries in an Adriamycin (ADR)-induced nephropathy rat model, which mimics human MCD or focal segmental glomerulosclerosis (FSGS).

## 2. Results

### 2.1. Expression of sEH in Renal Tissue of Patients with Primary Glomerular Diseases and Its Correlation with Clinical Indicators

Twenty patients with primary glomerular disease confirmed by renal biopsy were selected, including twelve males and eight females, with a mean age of 38.3 ± 10.3 years old. Among these patients, there were seven cases of IgAN, seven cases of MCD, and six cases of MN. In addition, five age-and-gender-matched renal cancer patients without other associated diseases were enrolled, and their surgically resected paracancerous tissues served as normal renal tissue controls. The characteristics of the patients are presented in Table 1.

The immunohistochemical analysis demonstrated that the expression of sEH in the renal tubules of patients with MCD (Figure 1C), IgAN (Figure 1D), and MN (Figure 1E) was significantly increased compared to those in the control group (Figure 1B), and some areas showed granular strong positive expression. The expression of sEH in renal tubules in patients with MCD and MN was slightly higher than that in patients with IgAN, but there was no statistically significant difference among the three groups (*p* = 0.263). In addition, sEH expression was also seen in the glomerular capillary endothelial cells of some patients, which was more obvious in the IgAN group (Figure 1D).

We further explored the correlation between the sEH expression level in renal tubules and the clinical indicators of the patients. The expression level of sEH was positively correlated with the 24 h urine protein excretion (r = 0.717, *p* = 0.006) and negatively correlated with serum albumin (r = −0.643, *p* = 0.023). The renal tubular sEH expression level has no correlation with age, systolic blood pressure, diastolic blood pressure, or serum creatinine (*p* = 0.935, 0.344, 0.293, and 0.563, respectively).

### 2.2. The Expression of Renal sEH Significantly Increased in Rats with ADRN

The qRT-PCR and WB results showed that the expression levels of renal sEH mRNA and protein were significantly increased in rats with ADRN. After the administration of the sEH inhibitor AUDA, there was no significant change in the expression levels of sEH. These results suggest that AUDA may exert its effects by inhibiting the catalytic activity of sEH without altering its expression, which is consistent with previous reports (Figure 2).

### 2.3. sEH Inhibition Reduced Urine Protein Excretion and Improved Renal Function in Rats with ADRN

We tested the blood and urine samples of rats from different experimental groups to evaluate the renal damage. The urine protein excretion significantly increased in rats with ADRN, accompanied by decreased serum albumin, increased blood lipids, and creatinine. Administration of the sEH inhibitor AUDA significantly reduced urine protein excretion and serum creatinine levels (Table 2), indicating the renal protective effect of sEH inhibition.

### 2.4. sEH Inhibition Alleviates Renal Pathological Damage in ADRN Rats

PAS staining and photography under a TEM were performed to observe the renal pathological changes in each group of rats. As shown in Figure 3, the rats with ADRN manifested with podocyte swelling and vacuolization in glomeruli (Figure 3A). Podocyte foot process effacement could be observed via electron microscopy (Figure 3B). Tubular injury was characterized by tubular atrophy, dilation, brush border loss, a large number of cast formations, and vacuolization of tubular epithelial cells (Figure 3A). The above pathological abnormalities were partially alleviated after the use of the sEH inhibitor.

### 2.5. The Effect of sEH Inhibitor on the Expression of Nephrin and Desmin in Glomerular Podocytes

We investigated the expression of nephrin and desmin using immunohistochemistry to further evaluate the function of the glomerular filtration barrier. Nephrin is an important molecule that forms the glomerular slit diaphragm, while desmin is a cytoskeletal protein in podocytes. The results showed that the expression of nephrin in glomerular podocytes in rats with ADRN was significantly downregulated, while the expression of desmin was significantly upregulated. More importantly, the sEH inhibitor AUDA markedly abrogated the abnormal expressions of nephrin and desmin induced by ADR (Figure 4). Collectively, these data suggest that sEH inhibition could prevent podocyte dysfunction and glomerular filtration barrier impairment, thus exerting renal protective effects.

### 2.6. sEH Inhibitor Suppressed Renal NF-κB Activation and Reduced TNF-α Levels in Rats with ADRN

We examined the effects of sEH inhibitor AUDA on NF-κB pathway activation in ADRN rats. As shown in Figure 5, the expression levels of p-IKKβ, p-IκBα, and p-p65 in renal tissue of rats with ADRN were significantly upregulated, indicating the increased activity of the NF-κB pathway. After inhibiting the effects of sEH, the phosphorylation of the above-mentioned proteins was markedly attenuated (Figure 5A–D), indicating that sEH inhibition can suppress the activation of the NF-κB pathway in renal tissue of rats with ADRN.

To further explore the effects of sEH inhibition on systemic and renal inflammation, we examined the levels of TNF-α both in the serum and kidney tissues in rats from different groups. The results showed that serum and renal TNF-α levels increased significantly in rats with ADRN. Administration of sEH inhibitor AUDA resulted in downregulated TNF-α in serum and kidneys (Figure 5E–G). Collectively, these data indicate that inhibiting sEH could mitigate the systemic and intrarenal inflammatory response in rats with ADRN.

## 3. Discussion

In this study, we demonstrated that renal sEH expression significantly increased in patients with MCD, IgAN, and MN. Renal sEH level was positively correlated with the severity of proteinuria and negatively correlated with serum albumin concentration. In an animal model of ADR-induced nephropathy, sEH was upregulated in the kidneys. Pharmacological inhibition of sEH suppressed renal inflammation and protected podocyte function, thereby reducing proteinuria and restoring renal function. Together, these findings suggested that sEH played an important role in the occurrence of proteinuria in primary glomerular diseases.

sEH is widely distributed in mammalian tissues but is particularly high in livers and kidneys. It converts EETs to their corresponding less potent diols, dihydroxyeicosatrienoic acids. A limitation of previous studies on sEH is that it was detected mainly in the kidneys of animal models. Little is known about sEH expression and location in human patients with kidney diseases. In the current study, we found that sEH expression levels in renal biopsy specimens were significantly upregulated in patients with glomerular diseases. sEH was preferentially located in proximal tubular epithelial cells, with relatively low levels in the vasculature. In addition, the renal sEH level was positively correlated with the severity of proteinuria in patients with glomerular diseases, which was consistent with the results of a previous report [9]. These data indicate that sEH might act as a notable participant in the pathogenesis of proteinuria in primary glomerular diseases. However, the number of patients included in this study is relatively small, and it should be validated in a larger population in the future to investigate the role of sEH in the pathogenesis of proteinuria. ADR-induced nephropathy results in nephrotic syndrome and podocyte injury, which is similar to human podocytopathies represented by MCD and FSGS. The experimental ADR-induced nephropathy rats in our study presented with severe proteinuria, widespread podocyte foot process effacement, tubulointerstitial injuries, and insufficient renal function. Importantly, renal sEH expression levels increased remarkably in rats with ADR-induced nephropathy. sEH inhibition reduced proteinuria, ameliorated histological damages, and preserved renal function. Previous results have shown that genetic disruption or pharmacological inhibition of sEH partially reversed albuminuria, morphological changes, and renal dysfunction in diabetic kidney disease and lupus nephritis [12,13,16,17,18,19]. Collectively, these findings point to a critical role of sEH in the onset of proteinuria in glomerular diseases. sEH can be considered a potential therapeutic target for the treatment of these disorders.

Podocytes are terminally differentiated epithelial cells located outside the glomerular basement membrane. The foot processes of neighboring podocytes were connected by a specialized membrane known as the slit diaphragm, which plays an essential role in preventing protein loss from the circulation. Nephrin has been proven to be a key molecule of the slit diaphragm. Desmin is a marker of podocyte epithelial–mesenchymal transition, a process resulting in podocyte cytoskeletal rearrangement and dysfunction [20]. Downregulation of nephrin and upregulation of desmin are observed in clinical cases and experimental models with several types of glomerular disease such as MCD [21], FSGS [22], MN [23], lupus nephritis [24], and diabetic kidney disease [25,26]. In our study, ADR-induced nephropathy manifested as extensive podocyte foot process effacement under electron microscopy, along with decreased expression of nephrin and increased expression of desmin. However, these alterations were partially reversed by an sEH inhibitor, indicating that the effects of sEH inhibition on reducing proteinuria were possibly attributed to the amelioration of podocyte impairment.

Immune-mediated systemic and intrarenal inflammation is considered to be an important pathway implicated in podocyte injury in glomerulopathies. The NF-κB axis is a key modulator of inflammatory responses. It regulates many proinflammatory cytokines involved in the pathophysiology of podocyte dysfunction and proteinuria. The pharmacological inhibition of NF-κB was found to ameliorate podocyte foot process effacement and proteinuria in an animal model of MCD [27]. Here, we confirmed the activation of inflammation in ADR-induced nephropathy, as evidenced by increased phosphorylation of p65-NF-κB and elevated TNF-α levels in kidneys and circulation. The administration of an sEH inhibitor markedly suppressed inflammation, thereby alleviating podocyte injuries and proteinuria. These results were supported by previous reports showing that podocyte sEH deficiency attenuated inflammation and subsequent renal damage in diabetic kidney disease and acute kidney injury [14,16]. Taken together, it may be reasonably speculated that sEH inhibition alleviates podocyte dysfunction and reduces proteinuria by mitigating inflammation.

## 4. Materials and Methods

### 4.1. Subjects

This study protocol was approved by the Peking University Biomedical Ethics Committee. Written informed consent was obtained from all participants. Patients confirmed by renal biopsy to have primary glomerular disease were enrolled. The exclusion criteria included patients with primary hypertension, diabetes, renal artery stenosis, heart failure, liver dysfunction, and malignancy. In addition, kidney cancer patients with no other comorbidities and matched age and gender were enrolled, and their surgically removed adjacent cancerous tissue was used as a normal control for renal tissue. Finally, 20 patients with primary glomerular disease and another 5 patients with kidney cancer were enrolled in this study. We collected clinical indicators such as age, blood pressure, urine protein, serum albumin, and serum creatinine from the included patients. Clinical parameters were measured by standard methods in our clinical laboratory.

### 4.2. Animal Modeling and Grouping

In this study, twenty-four male rats, at the initial age of 8 weeks, weighing 200–250 g, were obtained from Beijing Vital River Corporation. After 1 week of adaptive feeding, twenty-four rats were randomly divided into four groups: the control group (normal rats, n = 6), the AUDA group (normal rats administered with AUDA, n = 6), the ADR-induced nephropathy group (ADRN, n = 6) and the ADRN + AUDA group (ADRN rats administered with AUDA, n = 6). The ADRN rat model of glomerular diseases was established by two tail vein injections of Adriamycin. Following the method described in the literature, Adriamycin was prepared in a 2 mg/mL solution using injection water. The initial injection dose was 5 mg/kg, followed by another injection of 2.5 mg/kg one week later. The rats in the control group and AUDA group were injected with the same dose of sterile injection water via the tail vein. AUDA (Sigma Aldrich, St. Louis, MO, USA), an orally active sEH inhibitor, was given by gavage at 0.5 mg/(kg · d) for 4 weeks. Rats in the control and ADRN groups were administered equal amounts of saline. At the end of the experimental period, all rats were euthanized, and the kidneys were harvested. Blood and urine samples were collected. Serum creatinine (Scr), albumin (ALB), total cholesterol (TC), fasting blood glucose (FBG), alanine aminotransferase (ALT), and urine protein (UP) were measured using an automatic biochemical analyzer (Olympus AU 5400, Shinjuku City, Tokyo, Japan).

### 4.3. Quantitative Real-Time Polymerase Chain Reaction (qRT-PCR)

The transcript level of sEH was analyzed using quantitative real-time reverse transcriptase PCR (qRT-PCR). Briefly, total RNA was extracted with TRIzol reagent (Invitrogen, Waltham, MA, USA). RNA concentration and purity were assessed using a Nanodrop spectrophotometer (Thermo Fisher Scientific, Waltham, MA, USA). Then, a total of 1 µg of RNA was reversed-transcribed with M-MLV Reverse Transcriptase (Promega, Madison, WI, USA). Real-time PCR was performed using SYBR Green Mix reagent (Promega, Madison, WI, USA) with a Bio-Rad real-time PCR machine according to the manufacturer’s instructions. The transcript level of each mRNA was normalized by comparison with the mRNA of β-actin, and was calculated using the 2^−△△CT^ method. The sequences of primers of sEH used were forward, 5′-AAGCCTGTGGAGCCAGTCTA-3′, reverse, 5′-CCAGTTGTTGGTGACAATGC-3′.

### 4.4. Western Blot (WB)

Total protein was isolated from kidney tissues and quantified using the Bradford protein assay kit (Bio-Rad, Hercules, CA, USA). Equal amounts of proteins from kidney samples were separated by 10% sodium dodecyl sulfate–polyacrylamide gel electrophoresis (SDS-PAGE) and then electrophoretically transferred to a nitrocellulose membrane. Then, proteins were incubated with the following primary antibodies overnight at 4 °C: sEH (1:3000, Abcam, Cambridge, UK), phospho-IKKβ (1:1000, Abcam, Cambridge, UK), phospho-IκBα (1:1000, Cell Signaling Technology, Danvers, MA, USA), phospho-p65 (1:1000, Cell Signaling Technology, Danvers, MA, USA), and tumor necrosis factor alpha (TNF-α) (1:1000, ProteinTech, Wuhan, China). Horseradish peroxidase (HRP)-conjugated secondary antibodies (1:10000, Thermo Fisher Scientific, Waltham, MA, USA) were used to detect the binding of the primary antibodies. Protein bands were visualized using an enhanced chemiluminescence detection kit (Thermo Fisher Scientific, Waltham, MA, USA). Finally, the density of protein bands was counted using Image J software (version 1.80; NIH, Bethesda, MD, USA).

### 4.5. Detection of Serum TNF-α Levels by ELISA

The levels of serum TNF-α in each group of rats were measured by ELISA following the manufacturer’s instructions. To be specific, the obtained blood samples were centrifuged at a low temperature, and then the supernatant was collected. Each experimental condition was tested in three different wells and measured in duplicate. Lastly, the levels of TNF-α in the samples were calculated according to the established standard curves.

### 4.6. Immunohistochemical Examination and Morphological Observation

We used the immunohistochemical method to detect the expression of sEH in kidney tissues of patients with primary glomerular disease. For rats’ kidney tissues, we used periodic acid–Schiff (PAS) staining and a transmission electron microscope (TEM) to evaluate the morphological changes in the glomerular and cortical tubular areas. Meanwhile, we used immunohistochemical detection to observe the expression of nephrin and desmin in formalin-fixed, paraffin-embedded rat kidney sections. For immunohistochemistry, 3-micron-thick sections were deparaffinized and subjected to antigen retrieval. The sections were soaked in 3% H_2_O_2_ for 30 min to block endogenous peroxidase activity and then incubated overnight at 4 °C with the primary antibodies of sEH (1:250, Abcam, Cambridge, UK), nephrin (1:1000, Abcam, Cambridge, UK) and desmin (1:1000, Abcam, Cambridge, UK). Then, the slides were washed and added with HRP-labeled anti-rabbit secondary antibody (Thermo Fisher Scientific, Waltham, MA, USA) at room temperature for 30 min. DAB was added to the slides and stained with hematoxylin. Finally, the sections were photographed using a Leica microscope electronic imager, and a semi-quantitative analysis was performed. For TEM examination, the cortical regions of kidney tissues were sliced and minced into 1 mm^3^ pieces and fixed and prepared with the procedure described in the previous literature [28].

### 4.7. Statistical Analysis

Clinical data and biochemical indexes were shown as mean ± standard deviation. The data obtained were statistically analyzed using SPSS software, version 22.0 (IBM Corp., Armonk, NY, USA). Linear regression analysis was performed to explore the correlation between the expression level of sEH in renal tissue and clinical indicators of patients. For comparisons of groups, one-way ANOVA followed by a least square difference (LSD) multiple comparison test was used. Statistical significance was considered as *p* value < 0.05.

## 5. Conclusions

In conclusion, the current study identifies sEH as a contributor to podocyte dysfunction and proteinuria in glomerulopathies. sEH inhibition mitigates renal inflammation and podocyte injury, thus reducing proteinuria. These results extend previous findings of the reno-protective effects of sEH inhibition and suggest that sEH inhibition might be of therapeutic value in proteinuria.

## Figures and Tables

**Figure 1 ijms-25-10629-f001:**
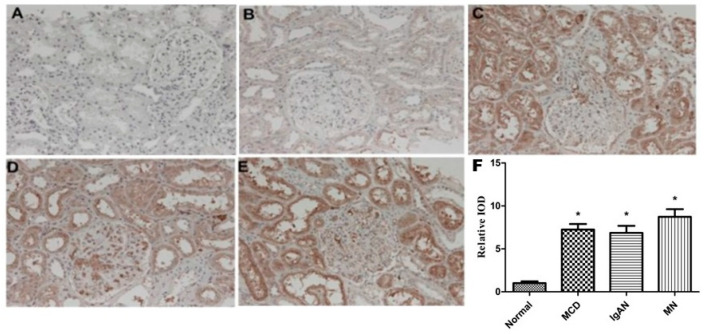
Expression of sEH in renal tissue. (**A**) Negative control; (**B**) normal kidney tissue; (**C**) kidney tissue of patient with MCD; (**D**) kidney tissue of patient with IgAN; (**E**) kidney tissue of patient with MN; (**F**) semi-quantitative results of immunohistochemistry. The image magnification is 400×. * *p* < 0.01 vs. normal group.

**Figure 2 ijms-25-10629-f002:**
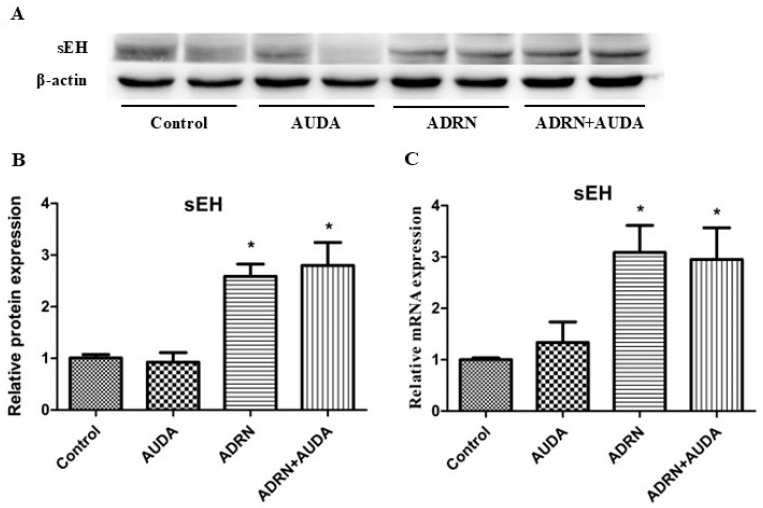
sEH expression increased significantly in rats with ADRN. (**A**) Representative WB images of sEH. (**B**,**C**) Quantitative analysis of the protein and mRNA expression levels of sEH in different groups. * *p* < 0.05 vs. control group.

**Figure 3 ijms-25-10629-f003:**
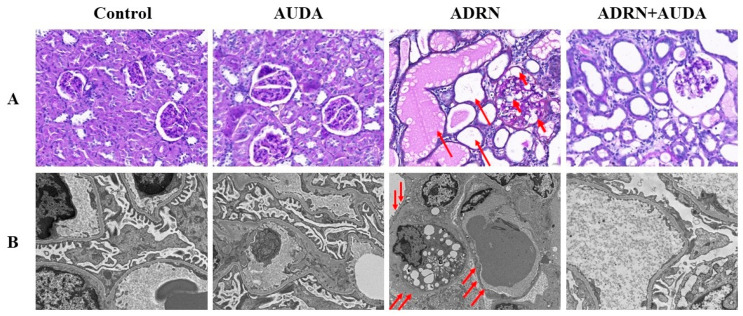
sEH inhibition alleviates renal pathological damage in ADRN rats. (**A**) Representative photomicrographs (PAS staining, magnification ×200) of the renal cortex from rats in different groups. Renal tissue was normal in control group and AUDA group. Renal pathological damage in rats with ADRN was characterized by podocyte swelling and vacuolization (short arrows), tubular dilation, and cast formation (long arrows). Pathological damages were alleviated in the ADRN+AUDA group compared to the ADRN group. (**B**) Morphological changes in the glomerular areas of the kidney samples under TEM (4200×). Glomerular morphology was normal in the control group and AUDA group. Diffuse podocyte foot process effacement (arrows) could be observed in the kidneys of rats with ADRN. Morphological changes were improved in the ADRN+AUDA group compared to the ADRN group.

**Figure 4 ijms-25-10629-f004:**
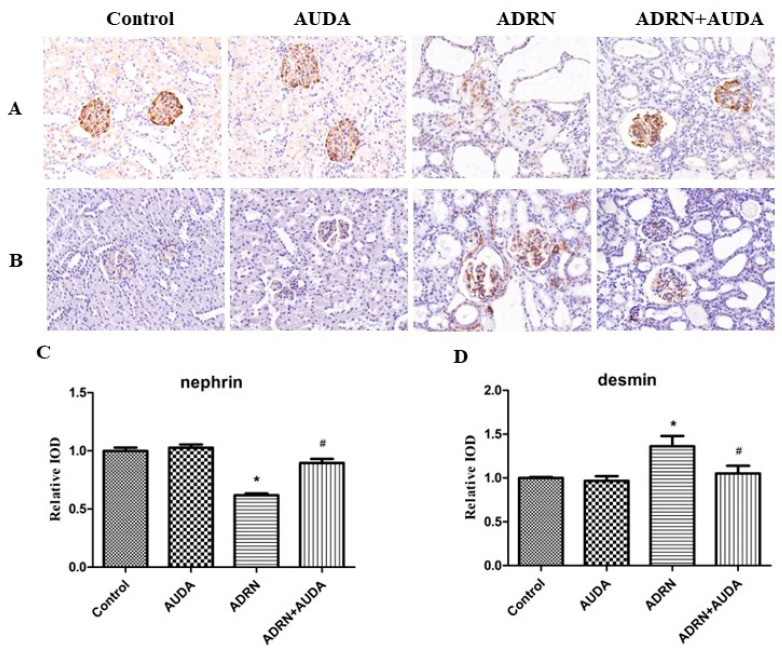
sEH inhibitor upregulated glomerular nephrin expression and decreased glomerular desmin expression in ADRN rats. (**A**) Representative images of nephrin immunostaining in kidney tissues (magnification ×200). Expression of nephrin was normal in control group and AUDA group, but significantly downregulated in ADRN group. Expression of nephrin in podocytes increased in ADRN+AUDA group compared with ADRN group. (**B**) Representative images of desmin immunostaining in kidney tissues (magnification ×200). Expression of desmin was significantly upregulated in ADRN group compared to control group, while sEH inhibitor AUDA suppressed desmin expression in podocytes. (**C**) Quantified graphs of nephrin. (**D**) Quantified graphs of desmin. * *p* < 0.05 vs. control group; # *p* < 0.05 vs. ADRN group.

**Figure 5 ijms-25-10629-f005:**
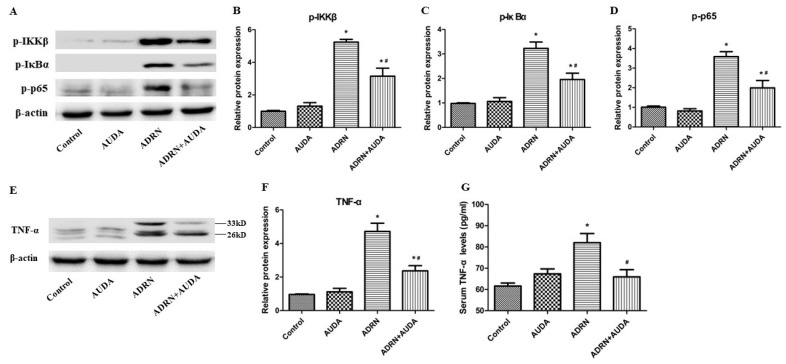
sEH inhibitor AUDA downregulated NF-κB activation and TNF-α levels in rats with ADRN. (**A**) Representative gel images of WB. (**B**–**D**) Quantified graphs of p-IKKβ, p-IκBα, and p-p65 in kidneys. (**E**) Representative gel images of WB. (**F**) Quantified graphs of TNF-α in the kidneys. (**G**) Serum TNF-α levels detected by ELISA. * *p* < 0.05 vs. control group; # *p* < 0.05 vs. ADRN group.

**Table 1 ijms-25-10629-t001:** The characteristics of the patients enrolled.

	Control (*n* = 5)	IgAN (*n* = 7)	MCD (*n* = 7)	MN (*n* = 6)
Age (years)	43.7 ± 16.5	36.4 ± 2.0	36.8 ± 11.1	42.2 ± 11.0
SBP (mmHg)	123.1 ± 4.9	127.6 ± 5.5	120.5 ± 6. 6	127.0 ± 6.7
DBP (mmHg)	75.2 ± 6.7	81.4 ± 6.3	78.8 ± 7.5	80.0 ± 7.1
UP (g/24 h)	0.116 ± 0.028	3.141 ± 1.850 *	6.205 ± 3.921 *#	9.529 ± 3.899 *#
ALB (g/L)	44.98 ± 4.41	39.92 ± 8.56	26.83 ± 10.13 *#	29.64 ± 7.00 *#
SCr (μmol/L)	58.72 ± 8.35	76.40 ± 13.95	63.25 ± 14.38	63.20 ± 13.91

ALB, albumin; DBP, diastolic blood pressure; IgAN, IgA nephropathy; MCD, minimal change disease; MN, membranous nephropathy; SBP, systolic blood pressure; SCr, serum creatinine; UP, urinary protein. * *p* < 0.05 vs. control group; # *p* < 0.05 vs. IgAN group.

**Table 2 ijms-25-10629-t002:** sEH inhibition attenuated renal damage in rats with ADRN.

	Control (n = 6)	AUDA (n = 6)	ADRN (n = 6)	ADRN + AUDA (n = 6)
UP (mg/24 h)	18.83 ± 4.56	13.44 ± 2.74	649.49 ± 53.86 *	489.76 ± 87.12 *#
ALB (g/L)	40.85 ± 2.96	40.35 ± 2.18	17.81 ± 2.67 *	19.63 ± 2.28 *
SCr (μmol/L)	26.67 ± 3.93	25.50 ± 3.83	123.50 ± 15.05 *	73.17 ± 8.98 *#
TC (mmol/L)	1.96 ± 0.21	1.99 ± 0.26	10.67 ± 1.87 *	10.50 ± 1.57 *
FBG (mmol/L)	9.39 ± 1.44	9.85 ± 1.20	8.51 ± 0.96	8.32 ± 2.71
ALT (U/L)	40.93 ± 6.48	39.17 ± 10.40	44.20 ± 6.61	36.25 ± 8.19

ADRN, Adriamycin-induced nephropathy; ALB, albumin; ALT, alanine aminotransferase; AUDA, a sEH inhibitor; FBG, fasting blood glucose; Scr, serum creatinine; TC, total cholesterol; UP, urine protein. * *p* < 0.05 vs. control group; # *p* < 0.05 vs. ADRN group.

## Data Availability

Data will be shared at the request of Yaoxian Liang (liangyaoxian@pkuph.edu.cn).

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
