# Peer review of "Soluble Epoxide Hydrolase Inhibition Attenuates Proteinuria by Alleviating Renal Inflammation and Podocyte Injuries in Adriamycin-Induced Nephropathy"

_ijms, 2024, doi:10.3390/ijms251910629_

Round 1

Reviewer 1 Report

Comments and Suggestions for Authors

My advice is rejection.

Given the limited sample size of 20 patients, how do you ensure that your findings are representative of the general population with primary glomerular diseases?
The study uses a small number of rats (24) in the experimental groups. Could you provide justification for the statistical power and how this sample size is sufficient to detect meaningful differences in your outcomes?
Could you please elaborate on the method of randomization that was employed to divide the animals into several groups? When gathering and evaluating data, did the investigators not know about the group assignments?
How can you explain the direct transfer of your findings to human settings in the absence of additional clinical evidence, considering that your findings about the therapeutic potential of sEH inhibitors are based on animal models?
How did you account for possible confounding factors such comorbidities, drug usage, and individual genetic variations in your human subjects?
Could you perhaps elaborate on the statistical methods applied in both human and animal studies to deal with possible outliers or missing data?
Patients with renal cancer provided the human control samples. How do you justify using these samples as controls, considering that cancerous and adjacent tissues might not represent healthy kidney physiology?
Previous studies have established the function of sEH in nephropathy. Could you describe your study's novel contribution in comparison to previous findings, particularly in the setting of primary glomerular diseases?
The control groups in your study were given either saline or water. Why didn't you include other controls, like rats treated with other recognized nephroprotective drugs, to provide a more complete comparison?
You propose that sEH inhibitors could be a viable therapeutic option for proteinuria in primary glomerular disorders. Given that your study does not include human clinical trials, what evidence supports this conclusion, and how do you intend to validate these findings in a clinical setting?

Comments on the Quality of English Language

Minor editing.

Author Response

Dear Reviewer 1:

Thank you very much for your advices. Thank you for your guidance in terms of professional knowledge. Your opinion is very instructive and needs to be considered in basic and clinical research. We have modified the article based on your suggestion, and answered your questions one by one. Pease see below for details. If there is any change that you think is inappropriate or does not meet your requirements, please point out that we will definitely modify it immediately.

Thanks again for your guidance, this is a great improvement in the quality of our articles. Answers see below:

  1. Given the limited sample size of 20 patients, how do you ensure that your findings are representative of the general population with primary glomerular diseases?

Our answer: The number of patients included in this manuscript is indeed relatively small. We just included 25 renal tissues (20 with primary glomerular diseases and 5 control), which can be seen as an exploration of the expression of sEH in the patients’ kidney, a preliminary experiment, rather than a definitive study. In this exploration, we discovered such a phenomenon that can be further studied with larger sample sizes in the future. But your advice is very reasonable, and we have added an explanation about this in the discussion section. As follows: ‘However, the number of patients included in this study is relatively small, and it should be validated in a larger population in the future to investigate the role of sEH in the pathogenesis of proteinuria.’ (see Discussion Para 2)

  1. The study uses a small number of rats (24) in the experimental groups. Could you provide justification for the statistical power and how this sample size is sufficient to detect meaningful differences in your outcomes?

Our answer: Before the formal experiment, we conducted a preliminary experiment and observed the differences between groups. And in previous studies, most of them used a design with 6 rats per group. Following the 3R principle (Reduction: The number of animals should be minimum number required to obtain statistically valid results) in animal experiments, we set a standard of 6 animals per group. Due to the limited length of the manuscript, we did not overly elaborate on the issue of rat sample size, only explaining the grouping method and quantity in the text. If you feel the need for further clarification, we can provide it as supplementary materials to further explain.

  1. Could you please elaborate on the method of randomization that was employed to divide the animals into several groups? When gathering and evaluating data, did the investigators not know about the group assignments?

Our answer: Purchase the same batch of rats with similar general conditions, and after one week of adaptive feeding, the animal laboratory staff will number the rats and randomly divide them into four through computer random numbers. The personnel responsible for testing blood and urine samples are laboratory technicians who are not aware of the grouping of rats, and the final data is generated by machines to ensure its objectivity. Then, the collection of other specimens and experimental processes are uniformly handled, to avoid inter group bias as much as possible.

  1. How can you explain the direct transfer of your findings to human settings in the absence of additional clinical evidence, considering that your findings about the therapeutic potential of sEH inhibitors are based on animal models?

Our answer: It has been reported that sEH inhibition could protect against acute kidney injury, diabetic nephropathy, lupus nephritis, hypertensive renal damage, tubular epithelial-mesenchymal transition and renal fibrosis. In recent years, our research group has been dedicated to studying the mechanism of action of sEH in kidney disease. Indeed, the process of translating the results of animal experiments into clinical practice is lengthy and requires multiple and multi-dimensional studies to summarize the findings before they can gradually be applied to clinical practice. sEH is expressed in both rat and human tissues, which is a verified fact. So, when we observe corresponding manifestations of kidney disease in rats, we want to verify this phenomenon in human kidney tissue. However, the study of mechanisms is lengthy and cannot be quickly applied to human research. Therefore, further research on the therapeutic potential of sEH inhibitors in humans should be conducted after the mechanism is clarified. In this article, only the expression of sEH was observed in human kidney tissue, and no intervention studies were conducted.

  1. How did your account for possible confounding factors such comorbidities, drug usage, and individual genetic variations in your human subjects?

Our answer: For the enrolled patients, we only collected clinical data at the time of disease diagnosis and observed the expression of sEH in kidney tissue. So, this is equivalent to baseline data, without the use of kidney protective drugs, which can reduce the impact of drugs. But it is true that comorbidities and individual genetic variations maybe need to be considered, especially when conducting treatment intervention studies in the future. At present, it is only an observation of the phenomenon. If we want to see the role played by different complications and genetic susceptibility factors, it may require further observation and subgroup analysis with a large sample size.

  1. Could you perhaps elaborate on the statistical methods applied in both human and animal studies to deal with possible outliers or missing data?

Our answer: For enrolled patients, we collected 6 variables of the clinical index, including age, blood pressure, urine protein, albumin and serum creatinine. For these variables, we evaluate the mean, distribution, maximum and minimum values, and also consider about the actual meaning of the values to determine whether there are outliers. In the animal experiment section, biochemical detection indicators include: urine protein, albumin, serum creatinine, total cholesterol, fasting blood glucose and alanine aminotransferase. When conducting descriptive analysis (including mean, quartile, maximum value, and minimum value), we only present real data and do not fill in missing data. If there is a missing data in regression analysis, then we Fill it in with the mean value.

  1. Patients with renal cancer provided the human control samples. How do you justify using these samples as controls, considering that cancerous and adjacent tissues might not represent healthy kidney physiology?

Our answer: Considering that it is difficult to obtain healthy kidney tissue (at that time, our hospital had not yet carried out kidney transplantation surgery), we could only try to find some tissues that could approach normal kidneys as much as possible. The kidney cancer postoperative patients selected in this article are all patients with normal renal function, no CKD, no other complications, and generally good condition. The tissue has been confirmed by pathologists and nephrologists as normal kidney tissue.

  1. Previous studies have established the function of sEH in nephropathy. Could you describe your study's novel contribution in comparison to previous findings, particularly in the setting of primary glomerular diseases?

Our answer: We agree that several studies have established the function of sEH in nephropathy. However, these studies mainly focused on the role of sEH in acute kidney injury, diabetic nephropathy, lupus nephritis, hypertensive renal damage, tubular epithelial-mesenchymal transition and renal fibrosis. Relatively little is known about the effects of sEH on primary glomerular diseases. In a previous study, Wang et al demonstrated that the level of sEH expression positively correlated with proteinuria and renal macrophages infiltration in IgA nephropathy (PMID: 29717935). However, the roles of sEH in primary glomerular diseases manifested as nephrotic syndrome, such as minimal change disease (MCD), focal segmental glomerulosclerosis (FSGS), and membranous nephropathy (MN), are currently poorly understood. In the present study, we investigated the renal sEH expressions in patients with primary glomerular diseases including MCD, IgA nephropathy, and MN. Furthermore, we studied the effects of sEH inhibition on renal inflammation and podocyte injuries in an Adriamycin-induced nephropathy rat model, which mimics human MCD or FSGS. We demonstrate for the first time that sEH inhibition mitigates renal inflammation and podocyte injury, thus reduces proteinuria in Adriamycin-induced nephropathy.

  1. The control groups in your study were given either saline or water. Why didn't you include other controls, like rats treated with other recognized to provide a more complete comparison?

Our answer: Thank you very much for your suggestion. In the present study, we want to verify whether the inhibition of sEH with AUDA effectively reduced urine protein excretion and attenuated renal pathological damage. If other medications are added for treatment, it is difficult to determine which medication is causing the aforementioned effects. So, in order to control for variables, we did not set up groups for other drug treatments in this study. We are considering designing another study to compare the efficacy of sEH inhibitors and RAS inhibitors in reducing proteinuria.

  1. You propose that sEH inhibitors could be a viable therapeutic option for proteinuria in primary glomerular disorders. Given that your study does not include human clinical trials, what evidence supports this conclusion, and how do you intend to validate these findings in a clinical setting?

Our answer: Thank you for your insightful question. In this study, we find that renal sEH expression was significantly increased in patients with primary glomerular diseases. Renal sEH level was positively correlated with the severity of proteinuria and negatively correlated with serum albumin concentration. Moreover, in rats with Adriamycin-induced nephropathy, sEH upregulated in the kidneys. Pharmacological inhibition of sEH suppressed renal inflammation and protected podocyte function, thereby reduced proteinuria and restored renal function. These results support the conclusion that sEH inhibition might be of potential value for the treatment of proteinuria in primary glomerular disorders. Because our study does not include human clinical trials, we could not be able to draw an unequivocal conclusion about the therapeutic value of sEH inhibition on proteinuria. Therefore, we used the words “might be”, “potential strategy” in the conclusion sections of the manuscript. Indeed, there have been several clinical trials evaluating the effects of sEH inhibitors on subarachnoid hemorrhage (PMID: 34873674), neuropathic pain (PMID: 33550801), hypertension (PMID: 29964123), and endothelial dysfunction in smokers (PMID: 27884766). We believe that in the near future there will be clinical trials to assess the effects of sEH inhibition on proteinuria in primary glomerular diseases, thus providing solid evidence on this issue.

Reviewer 2 Report

Comments and Suggestions for Authors

Qingyu Niu et al. analyzed the effects of soluble epoxide hydrolase on podocytes in various renal diseases and on urinary protein in adriamycin model animals with renal disease.

Soluble epoxide hydrolase is one of the substances that has been attracting attention in recent years, and this study, which focuses on podocytes, is considered to have great academic value.

1. Regarding the effects of soluble epoxide hydrolase on the kidney, how does it change with renal protective drugs such as ARBs, steroids, and SGLT2 inhibitors? If you have analyzed whether the effects of soluble epoxide hydrolase on podocytes differ when these renal protective drugs are administered in this study, please show us.

2. I think there is data evaluating soluble epoxide hydrolase in serum from cohort studies, etc. I think it would be better to show the data from clinical studies and discuss the differences with this study in more detail.

Comments on the Quality of English Language

n/a

Author Response

Dear Reviewer 2:

Thank you for your recognition and encouragement of our research, which is of great significance to us when we come here. Our team has been committed to basic and clinical research on primary glomerular disease and will continue to delve deeper into the pathogenesis and potential treatment methods of glomerular diseases in the future. We hope to have the opportunity to consult with you in the future. We have answered your questions. Please see below for details. If there is any change that you think is inappropriate or does not meet your requirements, please point out that we will definitely modify it immediately.

Thanks again for your guidance, this is a great improvement in the quality of our articles. Answers see below:

  1. Regarding the effects of soluble epoxide hydrolase on the kidney, how does it change with renal protective drugs such as ARBs, steroids, and SGLT2 inhibitors? If you have analyzed whether the effects of soluble epoxide hydrolase on podocytes differ when these renal protective drugs are administered in this study, please show us.

Our Answer: The question you raised is very meaningful and worth exploring. However, in this article, we included patients whose clinical indicators and kidney tissue were collected at the time of diagnosis, which is equivalent to baseline data. Therefore, we only investigated the sEH expression in patients with primary glomerular diseases who have not received renal protective medications. In the future, it is worth to explore the effects of renal protective drugs on soluble epoxide hydrolase expression, as well as whether the effects of soluble epoxide hydrolase on podocytes differ when these renal protective drugs are administered.

  1. I think there is data evaluating soluble epoxide hydrolase in serum from cohort studies, etc. I think it would be better to show the data from clinical studies and discuss the differences with this study in more detail.

Our Answer: These days, we have consulted relevant literature and found that there are almost no clinical studies measured the sEH level in serum and explored its relationship associated with kidney diseases. Most studies measure the sEH content in tissues. But there are also a few studies that measure the serum sEH content after applying SEH inhibitors to reflect the drug's effect. However, those studies are very different from the scientific objectives discussed in this article, so it is not appropriate to include them in the discussion of this article. So we have not yet added the contents of data evaluating soluble epoxide hydrolase in serum from cohort studies. We would like to asking for your advice again. If you think that it is not necessary to discuss the role of sEH in serum, then we will not add any relevant content. If you feel it is necessary, we will further search for literature and collect relevant results to add in the part of discussion. Thank you again for your suggestions.

Round 2

Reviewer 1 Report

Comments and Suggestions for Authors

The authors have provided sufficient and well-reasoned responses to most of the comments.